# Effects of Visual and Auditory Perturbations on Ski-Specific Balance among Males and Females—A Randomized Crossover Trial

**DOI:** 10.3390/ijerph16152665

**Published:** 2019-07-25

**Authors:** Martin Niedermeier, Elena Pocecco, Carolin Hildebrandt, Christian Raschner, Peter Federolf, Martin Kopp, Gerhard Ruedl

**Affiliations:** Department of Sport Science, University of Innsbruck, Fürstenweg 185, 6020 Innsbruck, Austria

**Keywords:** skiing, anterior cruciate ligament, environment, knee injuries, postural stability

## Abstract

Injuries in skiing show sex-specific differences, especially when visual perception is reduced. Reduced visual perception impairs balance, which plays an important role in avoiding skiing injuries. However, males and females might cope differently with reduced visual perception. Thus, the aim of this study was to evaluate sex-related effects of environmental perturbations (reduced visual perception and listening to music) on ski-specific balance. Using a crossover design, ski-specific balance was tested in 50 young adults (50% female) in four conditions: with and without listening to music and/or with and without reduced visual perception (ski goggles with occlusion foil). A four × two (condition by sex) mixed ANOVA revealed a significant condition by sex interaction, partial *η*² = 0.06. Females showed an increase in balance from the condition without music/with normal visual perception to the condition with music/with normal visual perception, while males showed a decrease. Balance was significantly higher in females compared to males, partial *η*² = 0.31. The findings suggest that balance is affected differently by environmental perturbations in females and males. However, the differences observed were not in line with our initial hypotheses, which might be because the model was too simplistic for how visual/auditory perturbations may affect balance.

## 1. Introduction

Alpine skiing is associated with a certain risk of injuries. Injury rates of one injury per 1000 ski days have been reported in alpine regions for recreational skiers [1,2]. The knee is the most affected injury location with 41% of all injury locations in Austrian skiers [2]. The prevalence of knee injuries shows a large sex difference: Females are about twice as likely to suffer a knee injury in skiing compared to males [1], and the knee is the most affected location in females [3]. Various variables including hormonal, behavioral, and neuromuscular factors have been discussed as explanatory factors for sex-specific differences in injury risk [3,4,5,6]. Another explanatory factor might be a sex-specific difference in coping with environmental perturbations. In this context, knee valgus moments during environmental perturbations (static laboratory skeleton imitating an opponent) were higher in females compared to males during sidestep cutting movements [7].

Environmental perturbations can be caused by other persons (e.g., body contact by another person), but also by conditions (e.g., reduced visual perception by weather conditions in outdoor sports). As more than 80% of all injuries in alpine skiing and snowboarding are non-contact in origin [2], conditions-driven environmental perturbations might play a major role in alpine skiing. A twofold prevalence of knee injuries compared to other injuries when skiing during snowfall was reported in females [8]. Visual perception is strongly impaired during snowfall, which might affect balance, as it is known that the visual sensory system is among the most important inputs for human postural control [9]. Poor balance can be considered a risk factor for sport injuries [10] and balance training was found to be protective against knee injuries in various sports [11]. In alpine skiing, being off-balance is one of the injury mechanisms [1]. Consequently, if males’ and females’ balance is differently affected by environmental perturbations, this might be a contributing factor for the explanation of sex-dependent knee injury rates and of the high prevalence of knee injuries during snowfall in females.

Another potentially relevant perturbation on ski-specific balance might be non-specific auditory perturbations, e.g., listening to music, which is quite popular during skiing. Listening to music has previously been connected to injuries in skiing and snowboarding; the odds for serious injuries were more than twice as high for snowboarders listening to music compared to snowboarders not listening to music [12]. Sex did not modify this observation. The percentage of females in those who listened to music was 10% and those who did not was 11% [12]. Injured slope tourers reported that they were listening to music during downhill skiing more often compared to non-injured slope tourers [13]. However, listening to music did not result in an increased reaction time to peripheral signals [14]. It might be possible that balance is affected by listening to music (instead of reaction time).

Thus, the aims of the present study were (a) to analyze possible sex-specific effects in coping with reduced visual perception on balance (postural stability) and (b) to analyze the effects of reduced visual perception and listening to music on postural stability. Based on the cited literature, we hypothesized that females’ postural stability in a reduced visual perception condition is affected more negatively compared to males’ postural stability. Furthermore, we hypothesized a negative influence of conditions with reduced visual perception and listening to music on postural stability for the total group.

## 2. Materials and Methods

### 2.1. Participants

The sample consisted of 50 healthy, physically active young adults (female: 50%). The participants studied Sport Science at the University of Innsbruck (convenience sampling) and mainly reported an advanced/expert skiing skill level (92%). The majority of the participants (52% of the total sample, female: 56%, male 48%) reported listening to music during exercising in general. No a priori power analysis was conducted; however, a sensitivity analysis using G*Power version 3.1 (University of Düsseldorf, Düsseldorf, Germany) [15] revealed that an effect size of partial *η*² = 0.07 was needed to be considered significant in the within–between interaction of a four × two mixed analysis of variance (assumptions: α = 0.05, power: 0.80, adjustment factor ε = 1). The study protocol was approved by the Institutional Review Board for Ethical Questions in Science of the University of Innsbruck (17.07.2012) in accordance with the Declaration of Helsinki 2008. Written informed consent was obtained from all participants before participating in this study.

### 2.2. Procedure

After information about the test procedure, self-reported demographic data were collected in a questionnaire with 22 items. Warm-up consisted of bipedal balance-specific exercises (five minutes) on a balance board (MFT Sport Disc, Trend Sport Trading GmbH, Austria) similar to the test device and of two trials on the test device for familiarization (i.e., the familiarization trials were conducted on the test day and prior to the first condition only). Using a crossover design, all participants were exposed to all experimental conditions in a randomized starting order. Balance was assessed for 30 s in four conditions with different environmental perturbations: without music/normal visual perception, with music/normal visual perception, without music/reduced visual perception, and with music/reduced visual perception. All participants wore standardized indoor shoes, ski poles (without touching the floor), an audio ski helmet, and ski googles. Music consisted of the song “Eye of the tiger” (Survivor) and was applied by the audio helmet (volume: 70–75 dB). An occlusion foil (Visus 0.3, resulting in a 70% reduction of visibility) was put on the ski google to create reduced visual perception (Figure 1). The standing position on the test device was identical in all conditions: bipedal, hip-wide stance with slightly flexed knees (knee angle: approximately 170°). A two-minute break was provided between the conditions. One balance test was conducted per condition. The total test duration was approximately 30 min for each participant.

### 2.3. Measurements

Balance (postural stability) was assessed using the Biodex Balance System (Biodex, Shirley, New York, NY, USA). The system contains a circular platform with a fixed point in the center, which allows multiaxial tilting of the platform. The participant’s task was to hold the platform levelled and as still as possible. A visual feedback about the position of the platform was provided by a small screen in front of the participant. An instable setting was selected (difficulty level 2 out of 8 levels with level 1 as the most unstable level). The displacement from the level position was recorded by the system and indices for medial–lateral stability (MLSI), anterior–posterior stability (APSI) and overall stability (OSI) was calculated [16]. Indices were expressed in degrees and higher indices marked a higher instability and thus a lower postural stability. Previous research reported acceptable inter-tester (intra-class correlation: 0.70) and intra-tester reliability (intra-class correlation: 0.82) for the overall stability index [16].

### 2.4. Statistical Analyses

All statistical analyses were performed using SPSS version 24 (IBM, New York, NY, USA). The primary analysis consisted of a four × two mixed analysis of variance (ANOVA), conducted to analyze the effect of the condition (without music/normal visual perception, with music/normal visual perception, without music/reduced visual perception, and with music/reduced visual perception), sex (female, male), and the condition by sex interaction on postural stability. A significant interaction between the within-subject factor condition and between-subject factor sex was considered as a sex-specific difference in coping with reduced visual perception and/or listening to music. Simple contrasts were used with “without music/normal visual perception” and “with music/reduced visual perception” as reference categories. OSI was used as the dependent variable. Two additional ANOVAs with MLSI and APSI as dependent variables were conducted to analyze if effects were driven by medial–lateral or by anterior–posterior stability.

In a secondary analysis, a possible habituation effect of listening to music in general during exercise was tested using a four × two mixed ANOVA on the dependent variable OSI. A significant interaction effect of listening to music in general during exercise (yes/no) and condition was considered as a different reaction to the conditions between those who listened to music in general during exercise and those who did not. Whenever the assumption of sphericity was not met, a Greenhouse Geisser correction was applied. Partial *η*² was used as an effect size with the classifications small (0.01), medium (0.06), and large (0.14) [17]. Data are presented as mean (SD). The level of significance was set at *p* < 0.05 (two-tailed).

## 3. Results

All 50 participants completed the study. Descriptive values are displayed in Table 1 for the total sample and by sex.

The analyses of stability indices by sex showed significant large-sized sex effects (Table 2). Females showed higher postural stability in all conditions and indices. No significant condition effect was found for any of the indices, indicating a comparable postural stability in all conditions. For OSI, a significant sex by condition interaction was evident, indicating sex-dependent different changes between conditions. Simple contrasts revealed that (a) females showed an increase in postural stability from the condition “without music/with normal visual perception” to the condition “with music/with normal visual perception”, where, on the contrary, males showed a decrease in postural stability. No significant contrast was found between the condition “without music/with normal visual perception” to the condition “without music/with reduced visual perception”.

Simple contrasts using the condition “with music/reduced visual perception” as a reference category revealed that (b) females showed a decrease in postural stability from the condition “with music/with normal visual perception” to the condition “with music/with reduced visual perception”, where males showed an increase in postural stability. APSI, but not MLSI, showed a significant condition by sex interaction indicating that interaction effects were mainly driven by anterior–posterior stability (and less by medial–lateral stability).

The secondary analysis on a possible habituation effect of listening to music in general during exercise showed no significant interaction, *p* = 0.816, partial *η*² < 0.01, indicating similar reactions to the different conditions between those who listened to music during exercise in general and those who did not.

## 4. Discussion

The main objective of the present study was to analyze possible sex-related differences on ski-specific balance in conditions with environmental perturbations (reduced visual perception and listening to music). While significant contrasts were found in the current study, these contrasts were not in line with the hypothesized sex-related differences. Contrary to the second hypothesis, no significant influence of environmental perturbations on postural stability was found for the total group. These results suggest on one hand, that females and males may respond differently to the environmental perturbations used in the present study, on the other hand, it might be an indication that our relatively simple model for how visual or auditory perturbations may affect postural stability is too simplistic. Other mechanisms, e.g., psychophysiological mechanisms such as arousal, may influence postural stability more than initially accounted for.

We hypothesized that females’ balance is affected more negatively in a reduced visual perception condition compared to males’ balance. However, this was not the case in the present study for the comparison of the conditions “without music/with normal visual perception” and “without music/with reduced visual perception”. Instead, the unexpected result was found that females’ balance was affected positively when listening to music compared to males’ balance, which was affected negatively. To the best of our knowledge, there is no study available for a sex-specific influence of listening to music on balance. However, there is evidence for sex differences in psychophysiological responses to musical stimuli [18]. Elevated response curves in skin conductance level were reported in females compared to males while listening to stimulating music [18]. One might conclude that this means a higher level of arousal in females, which—according to the Yerkes–Dodson Law [19]—might lead to a better performance in females. Although listening to music has been connected to injuries in skiing and snowboarding [12,13], it has to be mentioned that several authors discussed listening to music as potentially having a positive influence on motor performance abilities either through the rhythmical elements of the music [20] or by influencing the arousal [21]. The secondary analysis indicated that it is unlikely that the habituation effect of listening to music in general during exercise has to be considered as a confounding variable in the present study. Coping with the different conditions was similar between participants who reported listening to music during exercise in general and those who did not. Furthermore, the percentage of those listening to music during exercise in general was similar between sexes.

For the comparison of the condition “with music/with normal visual perception” and “with music/with reduced visual perception”, an increase was found in males’ balance, while females’ balance decreased. It was expected that males’ balance would also be negatively affected by reduced visual perception, as visual feedback is among the most important inputs on balance [9] and the absence of visual information (eyes closed) leads to a decrease in balance compared to eyes open [22]. Visual feedback was limited in the present study by wearing ski googles with an occlusion foil resulting in a 70% reduction of visibility; therefore, impaired balance was expected in the conditions with reduced visual perception for both sexes. However, this effect was not visible for males in the condition with reduced visual perception and music. It might be speculated that males’ balance is less affected by a multi-sensory impaired (visual and auditory) condition compared to females’. Indeed, impairments of balance in multi-sensory impaired conditions (reduced visual perception and somatosensory inputs) have been reported for females, while males coped more efficiently in multi-sensory impaired conditions [23]. Skiing can be considered a complex situation where many environmental influences potentially disturb skiers in maintaining balance. According to the present findings, males might be able to cope with multi-sensory environmental perturbations more efficiently compared to females. If this assumption holds true, sex-dependent differences in coping with environmental perturbations might be a contributing factor for the explanation of sex-dependent knee injury rates [1] and for the high prevalence of knee injuries during snowfall in females [8]. However, according to the present mixed results, we are not able to draw final conclusions regarding sex-specific effects in coping with reduced visual perception on balance.

As expected, higher postural stability was evident in females compared to males with a large-sized effect in overall, medial–lateral, and anterior–posterior stability. These findings are consistent with previous findings and can be explained by anthropometric factors (e.g., lower body height in females) [24], but also by neuromuscular, neurophysiological, and habitual factors (e.g., usage of higher heels in females) [25].

However, for the total group, postural stability in reduced visual perception conditions was only slightly lower compared to the condition without reduced perception and the condition effect was non-significant in the present study (small- to medium-sized effects). Several explanations for this finding can be considered. Firstly, the missing visual information might be compensated by other mechanisms, such as somatosensory and peripheral vestibular processing [9]. This compensation might be more important in the present sample compared to clinical samples or older participants, as the present sample consisted of physically active young adults, who are believed to possess better balance skills [24]. It was shown previously that subjects with better balance skills can cope more efficiently with missing visual information [26]. Secondly, the use of a feedback screen might have concealed the effect connected to impaired visual information. Despite the occlusion foil, subjects were still able to get (limited) visual information from the feedback screen. Visual information (even if limited) might be considered to be most relevant to maintain balance, as improved balance was reported when using a feedback screen compared to the condition without a feedback screen [22]. Thirdly and connected to the second explanation, the impact of the occlusion foil on balance might have been too small. The limited visual information was still enough to maintain comparable balance to the conditions without reduced visual perception. The occlusion foil was designed to imitate the environmental condition during snowfall. In a real skiing condition, however, reduced visual perception comparable to snowfall might have a larger influence on balance, as the situation is much more complex during skiing (e.g., moving skiers, no feedback screen available). Small sized-effects of environmental perturbations might accumulate in more complex, multi-sensory impaired situations [22]. Fourthly, outliers in the present data of the overall stability index might have led to an underestimation of the condition effect. An additional analysis not shown in the Results Section, excluding all potential outliers (defined as absolute z-standardized value > 1.96 [27], *n* = 9, all male), indicated a significant condition effect. The main effect of sex and the sex by condition interaction remained significant. However, this result must be interpreted cautiously because the exclusion of all potential outliers might be regarded as too conservative.

At least three limitations have to be considered when interpreting the findings. Firstly, a laboratory setting was used to control for confounding variables, which might be oversimplified, as the situation in skiing is much more complex compared to the setting used. The participants used ski poles, ski googles and a ski helmet, but did not wear ski boots. When wearing ski boots, balance is negatively influenced because of the restricted ankle joint [28], which might affect the results. Secondly, only one possible influencing source for knee injuries was investigated, i.e., affected balance. Although connected to injuries [1], a direct link between affected balance and injuries has not been proven yet. Thirdly, a group of volunteers with a high level of physical activity was studied, which might not be representative for the injured skiers in general.

## 5. Conclusions

The present findings suggest that balance is affected differently by environmental perturbations in females and males; however, the findings cannot provide clear evidence that females’ postural stability in a reduced visual perception condition (comparable to vision during snowfall) is affected more negatively compared to males’ postural stability. One possible interpretation of the results might be that psychophysiological mechanisms, such as level of arousal, may influence postural stability more than initially accounted for. Reduced sight (comparable to vision during snowfall) did not negatively influence balance in the present study, which might not reflect the complex situation of a real skiing situation (e.g., moving skiers, no feedback screen available). Future studies are needed to clarify the sex-specific effects of environmental perturbations on balance. These studies might use more complex, multi-sensory environmental perturbations to mimic the conditions on the ski slope more realistically.

## Figures and Tables

**Figure 1 ijerph-16-02665-f001:**
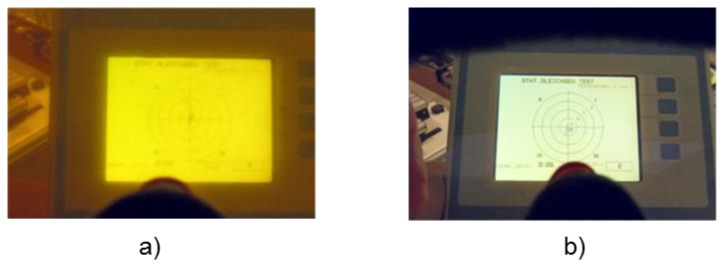
View on the visual feedback of the test device with (**a**) and without (**b**) occlusion foil.

**Table 1 ijerph-16-02665-t001:** Baseline characteristics of the study participants for the total sample and by sex.

	Total Sample (*n* = 50)				Female (*n* = 25)		Male (*n* = 25)	
	Mean	(SD) ^1^	Minimum	Maximum	Mean	(SD) ^1^	Mean	(SD) ^1^
Age (years)	23.5	(3.3)	18.0	33.0	23.1	(3.3)	24.0	(3.2)
Height (m)	1.73	(0.08)	1.50	1.87	1.68	(0.06)	1.78	(0.05)
Weight (kg)	66.8	(10.1)	46.0	92.0	59.4	(6.3)	74.3	(7.1)
Body mass index (kg/m^2^)	22.2	(2.2)	18.4	27.8	21.1	(2.0)	23.3	(1.7)

^1^ SD: standard deviation.

**Table 2 ijerph-16-02665-t002:** Mean (SD) postural stability indices by sex and condition.

		Condition								*p*-Value			η²p ^2^		
		Without Music and with Normal Visual Perception		With Music and With Normal Visual Perception		Without Music and With Reduced Visual Perception		With Music and With Reduced Visual Perception							
		Mean	(SD) ^1^	Mean	(SD) ^1^	Mean	(SD) ^1^	Mean	(SD) ^1^	Sex	Condition	Inter-Action	Sex	Condition	Inter-Action
Overall stability index [°]	female	2.25	(0.75)	1.97	(0.47)	2.25	(0.65)	2.31	(0.67)	**<0.001**	0.222	**0.022**	**0.31**	0.03	**0.06**
	male	3.28	(1.32)	3.79	(2.01)	3.86	(1.71)	3.53	(1.44)						
Anterior-posterior stability index [°]	female	1.74	(0.62)	1.48	(0.38)	1.78	(0.59)	1.82	(0.59)	**<0.001**	0.408	**0.029**	**0.27**	0.02	**0.06**
	male	2.43	(1.01)	2.9	(1.68)	2.82	(1.32)	2.61	(1.05)						
Medial-lateral stability index [°]	female	1.62	(0.49)	1.51	(0.38)	1.61	(0.37)	1.64	(0.44)	**<0.001**	0.151	0.108	**0.33**	0.04	0.04
	male	2.43	(0.86)	2.6	(1.22)	2.83	(1.24)	2.57	(1.04)						

Lower stability index values suggest a more stable postural control.^1^ SD: standard deviation, ^2^ η²p: effect size partial *η* squared. Bold values indicate significant *p*-values/effect sizes.

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
