# Peer review of "Effects of Visual and Auditory Perturbations on Ski-Specific Balance among Males and Females—A Randomized Crossover Trial"

_ijerph, 2019, doi:10.3390/ijerph16152665_

Round 1

Reviewer 1 Report

Overall, this is a clear and concise manuscript.

Line 30-48 - There are a lot of information about ACL injuries in the Introduction, not related to the subject. Please rephrase.

Minor English improvements are needed.

Author Response

Journal: IJERPH (ISSN 1660-4601)

Manuscript ID: ijerph-532365

Title: Effects of Visual and Auditory Perturbations on Ski-Specific Balance among Males and Females – A Randomized Crossover Trial

Authors’ response to the Reviewers

At first, we want to thank the Reviewers for their time spent on the evaluation of the manuscript. We believe the comments greatly helped to improve the manuscript.

Please find the comments of the Reviewers in black and the author’s responses in blue.

Reviewer #1

English language and style

( ) Extensive editing of English language and style required

( ) Moderate English changes required

(x) English language and style are fine/minor spell check required

( ) I don't feel qualified to judge about the English language and style

Yes      Can be improved        Must be improved      Not applicable

Does the introduction provide sufficient background and include all relevant references?           

                                                                                   ( )         (x)       ( )        ( )

Is the research design appropriate?                           (x)        ( )        ( )        ( )

Are the methods adequately described?                    (x)        ( )        ( )        ( )

Are the results clearly presented?                             (x)        ( )        ( )        ( )

Are the conclusions supported by the results?          (x)        ( )        ( )        ( )

Comments and Suggestions for Authors

Overall, this is a clear and concise manuscript.

Line 30-48 - There are a lot of information about ACL injuries in the Introduction, not related to the subject. Please rephrase.

Thank you for your comments. We rephrased the manuscript and focused less on anterior cruciate ligament injuries, but more on sex-specific differences in injuries in the lower extremities in general in the first part of the introduction. We eliminated most of the references referring to ACL injuries only.

The passage now reads as:

“The prevalence of knee injuries shows a large sex difference: Females are about twice as likely to suffer a knee injury in skiing compared to males [1] and the knee is the most affected location in females [7]. Various variables including hormonal, behavioral, and neuromuscular factors were discussed as explanatory factors for sex-specific differences in injury risk [6-8,11]. Another explanatory factor might be a sex-specific difference in coping with environmental perturbations. In this context, knee valgus moments during environmental perturbations (static laboratory skeleton imitating an opponent) were higher in females compared to males during sidestep cutting movements [9].”

Reviewer 2 Report

This study is about the effects of visual and auditory perturbations on ski specific balance tests. Moreover, it should be pointed out if there were any sex-specific differences. Against the hypotheses of the authors, the balance was higher in all four conditions compared to the male subjects.

Before publication, I have some points that should be addressed:

Line 17: Indicate in the abstract how many females and males participated in the study!

Line 39-40: Have you asked your female participants for ovulatory status? If I would speculate – if the majority of the female subjects were in the post-ovulatory phase, balance was higher and therefore this would have a positive effect on balance skills (since injury risk was reduced).

Line 62 to Line 69: Are there any sex-specific differences? If yes, (since your goal is to detect differences between males and females) please add them!

Line 81: You should look at your data if the remaining 48% had a different reaction to the conditions!

Line 90: Was there a familiarization session before the test day? Was there a familiarization with all four conditions?

Line 94: Was the balance test repeated within the same conditions?

Line 154: Since male subjects showed much higher standard deviations, you should exclude outliers (if there were any) and discuss the results at least in the discussion section!  

Line 231 to 233: I can imagine that this was the major reason for the unexpected results. Since females are commonly more flexible compared to males in a normal condition (without ski boots), females likely showed more balance due to their higher flexibility. This should be discussed in more detail in the discussion section!

Line 225: What is the common sport of the subjects in both females and males. There might be higher balance requirements in the commonly sports of your female subjects compared to the males?

Author Response

Journal: IJERPH (ISSN 1660-4601)

Manuscript ID: ijerph-532365

Title: Effects of Visual and Auditory Perturbations on Ski-Specific Balance among Males and Females – A Randomized Crossover Trial

Authors’ response to the Reviewers

At first, we want to thank the Reviewers for their time spent on the evaluation of the manuscript. We believe the comments greatly helped to improve the manuscript.

Please find the comments of the Reviewers in black and the author’s responses in blue.

Reviewer #2

English language and style

( ) Extensive editing of English language and style required

( ) Moderate English changes required

( ) English language and style are fine/minor spell check required

(x) I don't feel qualified to judge about the English language and style

Yes      Can be improved        Must be improved      Not applicable

Does the introduction provide sufficient background and include all relevant references?           

                                                                                   (x)        ( )        ( )        ( )

Is the research design appropriate?                           ( )         (x)       ( )        ( )

Are the methods adequately described?                    (x)        ( )        ( )        ( )

Are the results clearly presented?                             (x)        ( )        ( )        ( )

Are the conclusions supported by the results?          ( )         (x)       ( )        ( )

Comments and Suggestions for Authors

This study is about the effects of visual and auditory perturbations on ski specific balance tests. Moreover, it should be pointed out if there were any sex-specific differences. Against the hypotheses of the authors, the balance was higher in all four conditions compared to the male subjects.

Before publication, I have some points that should be addressed:

Line 17: Indicate in the abstract how many females and males participated in the study!

Thank you for this comment; we indicated in the abstract of the present version of the manuscript that sex distribution was 50% female. The sentence in the abstract now reads as:

“Using a crossover design, ski-specific balance was tested in 50 young adults (50% female) in four conditions:”

To fulfil the formal requirement of a maximum of 200 words, we slightly adapted two other sentences of the abstract.

Line 39-40: Have you asked your female participants for ovulatory status? If I would speculate – if the majority of the female subjects were in the post-ovulatory phase, balance was higher and therefore this would have a positive effect on balance skills (since injury risk was reduced).

We did not ask the female participants for ovulatory status. To the best our knowledge, it is relatively unclear, if ovulatory status affects balance abilities and contradictory findings are available: Abt et al. (2007) studied healthy and physically active females (21.4 ± 1.4 years) similar to the present study sample and reported no significant differences in postural stability between phases of the menstrual cycle. Sung et al. (2018) also tested young females (21.4 ± 1.4 years) and reported larger postural sway in limits of stabilities in the ovulation phase.

Furthermore, asking for ovulatory status might be considered a delicate/intimate question and might result in untrue answers and/or higher dropout rate.

Sung, E.S.; Kim, J.H. The influence of ovulation on postural stability (biodex balance system) in young female. J Exerc Rehabil 2018, 14, 638-642.

Abt, J.P.; Sell, T.C.; Laudner, K.G.; McCrory, J.L.; Loucks, T.L.; Berga, S.L.; Lephart, S.M. Neuromuscular and biomechanical characteristics do not vary across the menstrual cycle. Knee Surg. Sports Traumatol. Arthrosc. 2007, 15, 901-907.

Line 62 to Line 69: Are there any sex-specific differences? If yes, (since your goal is to detect differences between males and females) please add them!

If we understand the reviewer correctly, the reviewer is referring to possible sex-specific differences of listening to music, preferably in connection with injury rates. We are not aware of any information about sex-specific differences in listening to music. Russel et al. did not report sex-specific differences in snowboarders regarding listening to music with the percentage of females in those who listened to music with 10% and those who did not 11%. Sex distribution was not given in the other studies cited.

Since we believe that this information is also relevant for the reader, we want to thank the reviewer for the input and added the following passage to the manuscript:

“The odds for serious injuries were more than twice as high for snowboarders listening to music compared to snowboarders not listening to music [15]. Sex did not modify this observation. The percentage of females in those who listened to music was 10% and those who did not was 11% [15].”

Line 81: You should look at your data if the remaining 48% had a different reaction to the conditions!

Thank you for that comment. We re-analyzed the data using a 4 (condition) x 2 (listening to music during exercising in general yes/no) mixed ANOVA. No significant condition by listening to music during exercising in general (yes/no) interactions were found for the primary outcome OSI, p = .816, partial η² < 0.01, indicating similar reactions to the different conditions between those who listened to music during exercise in general and those who did not. Similarly, no significant main effects were found for listening to music during exercising in general, p = .505, partial η² = 0.01, indicating similar balance abilities between those who listened to music during exercise and those who did not.

We included the analysis to the manuscript and adapted the methods, results and discussion sections accordingly.

Methods:

“The majority of the participants (52% of the total sample, female: 56%, male 48%) reported to listen to music during exercising in general”

“In a secondary analysis, a possible habituation effect of listening to music in general during exercise was tested using a four × two mixed ANOVA on the dependent variable OSI. A significant interaction effect of listening to music in general during exercise (yes/no) and condition was considered as a different reaction to the conditions between those who listened to music in general during exercise and those who did not.”

Results:

“The secondary analysis on a possible habituation effect of listening to music in general during exercise showed no significant interaction, p = .816, partial η² < 0.01, indicating similar reactions to the different conditions between those who listened to music during exercise in general and those who did not.”

Discussion:

“The secondary analysis indicated that it is unlikely, that habituation effect of listening to music in general during exercise has to be considered as a confounding variable in the present study. Coping with the different conditions was similar between participants, who reported to listen to music during exercise in general, and those, who did not. Furthermore, the percentage of those listening to music during exercise in general was similar between sexes.”

Line 90: Was there a familiarization session before the test day? Was there a familiarization with all four conditions?

Two familiarization trials were conducted during the warm-up phase, which was on the test day. The warm-up phase was prior to the first condition only. We clarified this in the manuscript as follows:

“Warm-up consisted of (…) two trials on the test device for familiarization (i.e. the familiarization trials were conducted on the test day and prior to the first condition only).”

Line 94: Was the balance test repeated within the same conditions?

We did not use multiple tests per condition and clarified in the manuscript:

“A two-minute break was provided between the conditions. One balance test was conducted per each condition.”

Line 154: Since male subjects showed much higher standard deviations, you should exclude outliers (if there were any) and discuss the results at least in the discussion section! 

Thank you for the careful observation of the tables and the input. We checked the data for outliers following the procedure proposed in Field (2013). In detail, we z-transformed the primary outcome (OSI) in all four conditions separately and classified the absolute z-values in potential outliers (1.96<z<=2.58), probable outlies (2.58<z<=3.29), extreme scores (z>3.29) and normal range (all other values). Out of all values, 93% were in the normal range and 7% were at least potential outliers (please compare table below for all four conditions separately).

Without music and with normal visual perception

With music and with normal visual perception

Without music and with reduced visual perception

With music and with reduced visual perception

Extreme   (z-score>3.29) [%]

2

2

2

Probable   outliers (z>2.58) [%]

2

Potential   outliers (z>1.96) [%]

10

8

2

Normal   range [%]

88

90

98

96

Two additional analyses excluding outliers were performed the identical analysis (4x2 mixed ANOVA):

1) excluding probable outliers and extreme scores (excluded: n=3, all male) and

2) excluding all outliers and analyzing normal range only (excluded: n=9, all male)

According to analysis 1), main effects of sex, p <.001, partial η² = 0.31, and sex by condition interaction, p =.035, partial η² = 0.06, were found. Condition was not significant, p = .197, partial η² = 0.03.

According to analysis 2), main effects of sex, p .004, partial η² = 0.20, and sex by condition interaction, p =.040, partial η² = 0.07, were found. The condition effect emerged as significant, p = .016, partial η² = 0.08.

The additional analyses provide several insights: The main effect of sex and the sex by condition interaction is significant in all analyses. Discrepancy is present regarding a possible condition effect. In analysis 2), 100% of the values are considered in the normal range compared to the initial analysis (93% in the normal range) and analysis 1 (95% in the normal range). Although analysis 2) provides support for a possible condition effect, we believe that this analysis might be too conservative, since the initial analysis and analysis 1) correspond better with the expected value of 95% of the values in the normal range. Therefore, we followed the comment of the reviewer and cautiously discussed the analysis in the discussion section only without showing the analysis in the results section.

“Fourthly, outliers in the present data of the overall stability index might have led to an underestimation of the condition effect. An additional analysis not shown in the result section excluding all potential outliers (defined as absolute z-standardized value > 1.96 [31], n = 9, all male) indicated a significant condition effect. The main effect of sex and the sex by condition interaction remained significant. However, this result must be interpreted cautiously since the exclusion of all potential outliers might be regarded as too conservative.”

Field, A. Discovering statistics using IBM SPSS Statistics. 4th ed.; SAGE Publications: London, 2013.

Line 231 to 233: I can imagine that this was the major reason for the unexpected results. Since females are commonly more flexible compared to males in a normal condition (without ski boots), females likely showed more balance due to their higher flexibility. This should be discussed in more detail in the discussion section!

Thank you for that comment. We agree with the reviewer that flexibility is a relevant ability in certain balance tasks. However, we did not discuss the flexibility as a possible reason for the sex difference for the following reasons:

a) Higher flexibility is commonly used as an explanation for a better balance ability on tasks other than the Biodex Balance System (e.g. Y-Balance Test). Overmoyer et al. (2015) reported correlations between balance on the Y-Balance test and flexibility of the lower extremities of .5 to .8. However, in the present study, we used the Biodex Balance System, where the task was to hold the platform levelled and as still as possible. As such, minimal movement is involved compared to the Y-Balance Test. Therefore, we believe that higher flexibility might play a minor role in the present study with regard to the sex difference.

b) In previous studies using the Biodex Balance System, mainly anthropometric factors are discussed for the sex-specific difference (higher stability in females) (e.g. Ku et al. 2012, Greve et al. 2013). Greve et al. (2013) reported that anthropometric factors (body height, BMI, body weight) explain up to 72% of the variation in the overall stability index on the Biodex Balance System. Furthermore, they stated that “There is a consensus that the greater the height is, the worse the balance.” (Greve et al. 2013, p. 4).

c) The main aim of the present study was to analyze possible sex-specific effects in coping with reduced visual perception on balance and less on sex-specific differences in general. For that reason, we put the focus in the discussion more on the sex by condition interaction (and less on the well-known main effect of sex, when balance ability is assessed with the Biodex Balance System).

However, we are happy to adapt the manuscript, if the reviewer is not satisfied with our explanation.

Overmoyer, G.V.; Reiser, R.F. Relationships between lower-extremity flexibility, asymmetries, and the y balance test. The Journal of Strength & Conditioning Research 2015, 29, 1240-1247.

Greve, J.M.; Cug, M.; Dulgeroglu, D.; Brech, G.C.; Alonso, A.C. Relationship between anthropometric factors, gender, and balance under unstable conditions in young adults. Biomed Res Int 2013, 2013, 850424.

Ku, P.X.; Abu Osman, N.A.; Yusof, A.; Wan Abas, W.A. Biomechanical evaluation of the relationship between postural control and body mass index. J. Biomech. 2012, 45, 1638-1642.

Line 225: What is the common sport of the subjects in both females and males. There might be higher balance requirements in the commonly sports of your female subjects compared to the males?

We asked the participants for their preferred sport in the ski resort, which was similar between sexes: skiing (male: 70%/female: 76%), snowboarding (22%/16%) skitouring (9%/8%); discrepancies to the sum of 100% per each sex come from rounding. Unfortunately, information other than that on the general common sport is not available for the participants.

Round 2

Reviewer 2 Report

The authors addressed all my raised points.  Therefore, I would recommend to accept the manuscript in the current form.